# Stumbles, Gait, and Cognition: Risk Factors Associated with Falls in Older Adults with Subjective Memory Complaints

**DOI:** 10.3390/ijerph21121683

**Published:** 2024-12-17

**Authors:** Isabella Delgado, Miguel Angello Camacho, Isabella Pugliese, Hugo Juan Camilo Clavijo, Mabel Moreno, Beatriz Muñoz Ospina, Jorge Orozco

**Affiliations:** 1Centro de Investigaciones Clínicas, Fundación Valle del Lili, Carrera 98 No. 18–49, Cali 760032, Colombia; isadel999@gmail.com (I.D.); miguel.camacho.la@fvl.org.co (M.A.C.); mabel.moreno@fvl.org.co (M.M.); 2Faculty of Health Sciences, Universidad Icesi, Calle 18 No. 122–135, Cali 760031, Colombia; ipugliese03@gmail.com (I.P.); hugojuancamilo@gmail.com (H.J.C.C.); 3Adults Neuropsychology, Fundación Valle del Lili, Carrera 98 No. 18–49, Cali 760032, Colombia; 4Department of Human Sciences, Universidad Icesi, Calle 18 No. 122–135, Cali 760031, Colombia; 5Neurology Department, Fundación Valle del Lili, Carrera 98 No. 18–49, Cali 760032, Colombia; jorge.orozco@fvl.org.co

**Keywords:** falls, aging, memory disorders

## Abstract

Falls are a public health problem, impacting quality of life, independence, and health costs. Subjective memory complaints (SMCs) and mild cognitive impairment (MCI) increase with age and may coexist. The risk of falls coinciding with SMCs is less understood. This study explored the risk factors associated with falls in adults with SMCs or MCI. A case–control study in adults over 50 was conducted. All participants underwent a neuropsychological assessment and a Timed Up and Go (TUG) test for gait analysis. Logistic regression calculated OR and *p* values, adjusting for demographic, clinical, cognitive, and gait variables. There was a total of 64 patients (47.06%) and 72 controls (52.94%). Fallers were older (70.76 ± 7.31) and had hypertension (29.63%), a history of stumbling (13.97%), slow TUG test performance (19.12%), and an asymmetric arm swing (19.85%). Fallers had lower verbal fluency (*p* = 0.043) and impairment on the Rey–Osterrieth figure copy (*p* = 0.047). Highest risk factors included female sex (OR 3.55, *p* = 0.006), older age (OR= 1.08, *p* = 0.006), hypertension (OR 3.33, *p* = 0.005), and stumbles (OR 5.65, *p* = 0.002). This study reconsiders clinical fall risk assessments in older adults with SMCs. Visuo-constructional abilities and executive dysfunction should be followed over time. Female sex, hypertension, and stumbles are risk factors. Integrated cognitive and motor assessments are crucial for detecting and proposing interventions for fall prevention in this population.

## 1. Introduction

Falls in adults over 65 years of age constitute a serious public health problem due to their negative impact on quality of life, the direct and indirect costs to healthcare system, and caregiver burden [1]. Falls are the second leading cause of unintentional deaths worldwide. The risk factors for falls can be classified as intrinsic or extrinsic. Intrinsic factors are related to the individual and include older age, medication side effects, physical inactivity, chronic health conditions, cognitive impairment, and dementia [2,3]. Extrinsic factors involve environmental characteristics, such as inadequate lighting and uneven surfaces [4].

Subjective memory complaints (SMCs) are defined as the subjective perception of cognitive decline compared to previous levels of functioning in individuals with normal cognition [5]. Subjects with SMCs experience subjective problems or changes in their memory, which can occur in those with mild cognitive impairment (MCI) as well as in people without any objective cognitive decline [6]. SMCs tend to increase with age and are common among older adults. Brigola et al., found that approximately 50% of people aged 50 to 59 experience SMCs, and there is an association between SMCs and cognitive impairment [7]. This is according to Palmer et al., who found that 51% of subjects who developed dementia had SMCs at baseline [8]. Multiple studies indicate that individuals with SMCs are at a higher risk of developing MCI and dementia; however, the reported conversion rates vary significantly between studies [9,10,11,12]. Accordingly, SMCs constitute an important domain of study beyond cognitive impairment and Alzheimer’s disease. Research into SMCs could facilitate the development of early detection methods to identify individuals at risk, and to explore motor dysfunctions related to Alzheimer’s disease [13].

Gait disturbances often coexist in older people. Slow gait speed has been associated with mild cognitive impairment and is a predictive factor for the development of dementia [14]. However, the mechanisms involved in the association between gait and cognition have not been completely determined. Evidence indicates that visual attention and executive function are strongly associated with gait dysfunction and the occurrence of falls [15,16,17,18]. Executive function (EF) is a group of processes that include attention, working memory, inhibitory control, and cognitive flexibility, all of which are essential for walking. The crucial role of EF in walking involves processing speed, which is associated with gait speed, the double support phase, and maintaining balance during walking [16]. Additionally, EF might also contribute to the ability to recover from extrinsic factors [17]. Research into the relationship between cognition and gait is ongoing. Indeed, based on current findings regarding attentional processes and executive function, it is reasonable to consider that future studies of fall risk could include an evaluation of these cognitive domains.

Recently, a clinical predementia syndrome known as cognitive–motor risk syndrome has been described, characterized by a combination of a reduction in walking speed and self-reported cognitive complaints [14]. In Colombia, based on the SABE study, the prevalence of this syndrome was found to be 10.7% of older adults and was associated with older age, low education, lower Minimental State Examination (MMSE) scores, and various chronic conditions, including mental health disorders, a history of myocardial infarction, hypertension, and diabetes [19].

Multiple studies have established a link between dementia and MCI with a higher risk of falling [15] but less is known about the relationship between SMCs and the occurrence of falls in older people. Although studies of risk factors associated with falls in Colombia have provided valuable knowledge, there are few studies regarding adults aged 50 years and older in the Colombian population describing risk factors associated with falls. This study aims to investigate the risk factors associated with falls in a Colombian population aged over 50 years with SMCs. This is the first study that includes a comprehensive neuropsychological assessment as part of the evaluation protocol for falls in this population. Identifying the risk factors and cognitive domains associated with falls in this population can lead to the improvement of our fall prevention strategy in older adults.

## 2. Materials and Methods

### 2.1. Study Design and Population

An analytical observational case–control study (1:1) was conducted from a cohort of participants between September 2018 and December 2022 from the Hospital Universitario Fundación Valle del Lili in Cali—Colombia. Participants reporting SMCs were conveniently recruited from the adult Neuropsychology Service. The identification and selection of participants in this study were based on a review of digital medical records of individuals who reported subjective memory complaints, regardless of whether any degree of cognitive impairment was detected through objective neuropsychological assessment. The subjects were divided into two groups, cases (fallers) and controls (non-fallers), based on the self-reported occurrence of a fall in the past year. Age was stratified into two groups: <65 years and >65 years of age. Falls were defined as unintentional events that resulted in a person reaching the ground, floor, or other lower level [15]. The Institutional Review Board of Fundación Valle del Lili, Cali, Colombia, approved the study. This work was conducted according to the Helsinki Declaration. Written informed consent was obtained from all subjects (cases and controls).

The inclusion criteria for this study were adults aged 50 years and older who attended the Neuropsychology Service and reported SMCs. The exclusion criteria for the study were subjects with a history of benzodiazepine or antidepressant consumption because these drugs can contribute to falling by causing sedation, impaired balance, and orthostatic hypotension [15]. In addition, subjects with movement disorders, a previous diagnosis of dementia or epilepsy, a disability that could affect walking independently, or the use of walking aids were also excluded. A case was defined as a subject over 50 years old with SMCs who reported at least one fall, without a resulting fall-related injury (any noticeable physical harm or pain) [15] in the 12 months before the neuropsychological assessment. A control was defined as a subject over 50 years old with SMCs without falls in the 12 months prior to the neuropsychological assessment.

The Timed “Up & Go” (TUG) test is a reliable and valid test for quantifying functional mobility that may also be useful in following clinical change over time. The test is quick, requires no special equipment or training, and is easily included as part of routine medical examination. Single-task and cognitive dual-task TUG tests were instructed for all subjects [20].

Sample size estimation was performed considering minimal cognitive impairment as a risk factor for falls. Based on the prevalence of cognitive impairment in subjects with falls (64.8%) and an odds ratio showing a 3.28 times greater likelihood of falls in the exposed group compared to the non-exposed group [21], and with a case-to-control ratio of 1:1, a 95% confidence level (α 0.05), and a statistical power of 90% (β 0.20), a minimum of 142 participants (71 cases and 71 controls) was required. Sample size calculation was performed in OpenEpi version 3.01 2013 free distribution.

### 2.2. Neuropsychological Assessment and Falls

For this article, these terms were defined as follows:-Stumble: an event where the person almost fell but was able to catch himself/herself or stop the fall [22]. Evaluated by presence or absence.-Instability: loss of balance or unsteadiness while walking or feeling imbalanced. Evaluated by presence or absence.-Arm swing asymmetry: reduced coordination and active arm retroversion on one side compared to the other [23,24]. Evaluated by presence or absence.-Unsteadiness during TUG testing: the subject demonstrated all 3 characteristics, a slow tentative pace, loss of balance, and reduced or no arm swing during the trials, observed by the evaluator [20]. Evaluated by presence or absence.-Single-task TUG: obtained in accordance with the protocol and the reference values for the single-task Timed Up and Go [25]-Dual-task TUG: Timed Up and Go test performed with the addition of a secondary cognitive task.-Cognitive cost: in gait, cognitive cost quantifies the additional demand on cognitive resources required when performing a dual task. It is calculated as the percentage difference between the estimated single-task and estimated dual-task performance [26] using the following formula:
$$\mathrm{cognitive}\;\mathrm{cost}=\frac{\mathrm{Single}\;\mathrm{task}\;\mathrm{condition}-\mathrm{dual}\;\mathrm{task}\;\mathrm{condition}}{\mathrm{Single}\;\mathrm{task}\;\mathrm{condition}}\times 100$$

-Estimated single-task walking speed: the gait speed is estimated based on the single-task TUG by dividing the 6 m distance by the total time, including the standing, walking, turning, and sitting phases. Evaluated in m/s.-Estimated dual-task walking speed: the gait speed is estimated based on the dual-task TUG, by dividing the 6 m distance by the total time, including the standing, walking, turning, and sitting phases. Evaluated in m/s.

A trained neuropsychologist evaluated all the participants. An interview was conducted to explore the participant’s medical history, history of stumbles with/without falls in the previous 12 months, and functionality in basic and instrumental activities of daily living; and a comprehensive neuropsychological evaluation was also conducted, including the Montreal Cognitive Assessment (MoCA) with a global cut-off point of less than 22/23 suggesting MCI [26]. The neuropsychological battery consisted of at least 1 test in 3 domains: (1) visuo-constructional abilities were evaluated with cube copying and a copy of the Rey–Osterrieth Complex Figure Test; (2) language and semantic knowledge were evaluated by assessing semantic verbal fluency, and (3) executive functions were evaluated using the Frontal Assessment Battery (FAB) and phonemic fluency. To complement the assessment, arithmetic and digit span tasks of the Wechsler Adult Intelligence Scale Fourth Edition (WAIS-IV) were used to calculate the working memory index (WMI). The recall of the Rey–Osterrieth Complex Figure Test was used to evaluate non-verbal memory. Logical Memory and Verbal Paired Associates of the Wechsler Memory Scale Fourth Edition (WMS-IV) were used to calculate the auditory memory index.

Gait was evaluated with the Timed Up and Go test (TUG) [25]. Participants were instructed to rise from a standard chair with their arms, walk three meters, turn, walk back to the chair, and sit down again. Subsequently, participants performed a dual-task TUG (TUGDT) [27], which involved completing the TUG while simultaneously performing a serial subtraction task (subtracting 3 starting from 20). The time was recorded with a stopwatch. All the subjects executed two trials for each condition. The time required to complete each task was measured and used for evaluation. The mean time of the two trials for each condition was calculated and two values for the TUG and TUG dual tasks were obtained. These values were used to calculate the cognitive cost (CC) and were included in the statistical analysis. All trials of the TUG were video-recorded, and gait characteristics were assessed and described.

### 2.3. Statistical Analysis

A descriptive analysis of the sociodemographic characteristics, clinical history, gait evaluation, and neuropsychological assessment of the cases and controls was conducted. The normality of each of the numerical variables was evaluated with the Kolmogorov–Smirnov test. Numerical variables with a parametric distribution are reported as the mean and standard deviation, whereas those with a non-parametric distribution are reported as the median and interquartile range. Categorical variables are presented as absolute and relative frequencies. For continuous quantitative variables, the Student’s *t* test for independent samples or the Mann–Whitney U test was used. The Pearson’s chi-square test or the Fisher’s exact test were used for categorical variables to find significant differences between groups.

To identify the factors associated with the risk of falls in older subjects with SMCs, a multivariate analysis was performed using a logistic regression model to calculate the odds ratio (OR), 95% confidence intervals (95% CI), and the *p* value. According to the World Health Organization, people over 65 years old are at a higher risk of falling and suffering fatal falls. For this reason, we explored the importance of participants’ age by stratifying the variable into two groups: <65 years and ≥65 years. We evaluated the relationship of each risk factor across the two age groups using descriptive and bivariate analysis. To identify risk factors associated with falls, variables with a statistical significance of <0.20 obtained in the simple logistic regression model were selected for entry into the multiple regression model. The backward elimination procedure was used until the most parsimonious model was obtained. After adjusting for demographic variables, medical history, gait characteristics, and cognitive assessment, variables with a *p* value of <0.05 were considered. The goodness-of-fit of the final model was evaluated using the Hosmer–Lemeshow test (H-L). A *p* value of <0.05 suggests that the model accurately predicts the observed values. The ability to discriminate between falling and non-falling patients was assessed by the area under the ROC curve. If the area under the curve is greater than 0.7, it can be concluded that the final model has acceptable discriminatory ability. The statistical package STATA ^®^ V 18.0 (College Station, TX 77845, USA) for Windows was used.

## 3. Results

### 3.1. Demographic Characteristics and Clinical History

The clinical characteristics of individuals who fall (cases) and those who do not fall (controls), including gait characteristics, are presented in Table 1. Most participants were women (60.29%), were older than 65 years, and had more than 12 years of education (technical and professional (50.74%) and postgraduate (14.71%)). The analysis between the groups showed that the cases (fallers) were significantly older (70.76 ± 7.31 vs. 64.55 ± 7.95; *p* ≤ 0.001), had a lower level of education, had a higher frequency of arterial hypertension (29.63% vs. 16.30%, *p* < 0.001), and more frequent occurrences of stumbles (13.97% vs. 3.68%, *p* = 0.001) compared to the controls (non-fallers). Furthermore, among participants who consumed antihypertensive drugs, more cases than controls reported taking antihypertensive medications (68% vs. 27.91%, *p* = 0.001), particularly among those consuming a single drug. Cognitive screening using the MoCA test showed that both groups (cases and controls) had MCI without statistically significant differences (23 (IQR 20–25) vs. 23 (IQR 21–26), *p* = 0.349). No significant sex differences were found between groups, as shown in Table 1.

### 3.2. General Motor and Cognitive Performance: Differences Between Groups

The cases group needed longer time (in seconds) to complete the single-task and dual-task TUG test ((mean ± SD) 14.43 ± 3.48 vs. 13.01 ± 2.69 and 17.24 ± 5.07 vs. 15.00 ± 3.90), respectively, with significantly slower estimated single-task and dual-task walking speeds (m/s) ((Mean ± SD) 0.43 ± 0.09 vs. 0.47 ± 0.08 and 0.37 ± 0.10 vs. 0.42 ± 0.10), respectively, and were more likely to have unsteadiness as observed by the examiner during TUG testing (9.56% vs. 2.94%, *p* = 0.009), and asymmetric arm swing testing (19.85% vs. 13.24%, *p* = 0.033) compared to the control group. Additionally, the neuropsychological assessment exhibited lower phonological verbal fluency (10.53 ± 3.62 vs. 11.81 ± 3.29 *p* = 0.043) and poorer performance on the copy of the Rey–Osterrieth Complex Figure Test (*p* = 0.047) in the cases compared with the controls. However, other cognitive domains, including nonverbal and verbal memory, semantic fluency, and executive functions, did not show significant differences between groups, as shown in Table 2.

### 3.3. Effect of Age Among Risk Factors

The effect of age and the different risk factors are summarized in Table 3. As age increases, the frequency of chronic diseases tends to rise, such as arterial hypertension (50.55%), dyslipidemia (36.26%), hypothyroidism (39.56%), and cardiovascular diseases (92.31%). Additionally, age exacerbates abnormalities in gait characteristics, including postural instability (71.43%) and a history of stumbles (22%). Participants aged 65 years or older needed longer time (in seconds) to complete the single-task and dual-task TUG task (mean ± SD) (12.65 ± 2.24 vs. 14.19 ± 3.42) and (14.55 ± 3.81 vs. 16.80 ± 4.80), respectively. Older participants also had significantly slower estimated single-task and dual-task walking speeds (m/s) (mean ± SD) (0.48 ± 0.08 vs. 0.44 ± 0.09) and (0.43 ± 0.10 vs. 0.38 ± 0.09). No significant difference was found in cognitive cost.

### 3.4. Fall Risk Profile

The risk factors associated with falls are shown in Table 4. In the simple model, female sex (OR = 1.96, *p* = 0.059), age (OR: 1.11, *p* < 0.001), a postgraduate degree (OR = 1.46, *p* = 0.463), arterial hypertension (OR = 3.71, *p* < 0.001), and stumbles (OR = 5.65, *p* = 0.001) were risk factors for falls. After model adjustment by confounding variables, the results showed that female sex, age, having a history of arterial hypertension, and stumbles were associated with falling. The final model also revealed that higher education levels were associated with a significant decrease in the risk of falls (OR: 0.36, *p* = 0.021). Based on the goodness-of-fit of the final model, we can suggest that the model fits the observed data (*p* value: 0.2432). Furthermore, the adjusted model demonstrated acceptable discriminatory capacity (AUC: 0.8153) to identify patients with mild cognitive impairment and probability of falling (Figure 1).

According to the criteria used to calculate the theoretical sample size, which defined a minimum of 142 participants, and the sample size achieved in this study of 136 participants, the statistical power was calculated to identify potential Type I errors (false positives) and Type II errors (false negatives) in the estimates. Based on the premise of our alternative hypothesis that the probability of falls in individuals with mild cognitive impairment is higher than reported by the study of Monachan et al. [21], the power was calculated using the calculator (^1^) with the following parameters: the baseline probability under the null hypothesis (p0 = 0.648) [21]; the probability under the alternative hypothesis (p1 = 0.781) (obtained in our study); the alpha level (α = 0.05); the effect size (R² = 0.0034); and the sample size (n = 136). The result was a power of 87%.

## 4. Discussion

This study investigated the risk factors associated with falls in a Colombian population aged over 50 years with SMCs. The results revealed that most of the participants were women, older than 65 years of age, and had more than 12 years of education. In the study, elderly participants were identified with mild cognitive impairment (MCI) based on their MoCA scores, with a mean score of 23 (IQR 21–25). Subgroup analysis by age (<65 years vs. >65 years) revealed that single-task and dual-task TUG performance significantly differentiated between fallers and non-fallers. The estimated occurrence of stumbles approached statistical significance (*p* = 0.060).

**Those in the cases group were older, with a higher proportion reporting a history of stumbles and arterial hypertension:** Our results confirm that falls were more frequent in the older population of 65 years old and above. These factors can be attributed to changes in the musculoskeletal and neuromuscular systems, leading to impaired balance and increased risk of falls [28,29]. Age may also be associated with frailty, polypharmacy, and comorbidities, all widely recognized as risk factors for falls [15]. In addition, aging may be associated with a more sedentary lifestyle, which increases the probability of unsafe walking and then falling. In medical history, arterial hypertension was significantly higher in the cases group.

Patients with arterial hypertension have a higher risk of falling due to the use of some antihypertensives, such as diuretics [30]. Moreover, based on the SABE study results, Marquez et al., evaluated cognitive–motor risk syndrome and found that these factors are related to a slower walking speed and several chronic conditions, such as a history of myocardial infarction, mental disorders, diabetes, and arterial hypertension [19]. There is evidence of several chronic health conditions that could be associated with falls, such as cardiac disease, arterial hypertension, and diabetes [31]. Our study found that arterial hypertension increases the risk of falling up to 3.33 times compared to those without arterial hypertension. However, new studies with a larger sample size are required to explore the relationship between falls and pathological conditions and the use of antihypertensives in the Colombian population.

On the other hand, our cases group had a high frequency of stumbles, which can be related to an impaired corrective response to extrinsic factors during walking and a greater loss of confidence in the ability to walk and maintain normal balance [32], consequently leading to a higher risk of falling. Previous studies have indicated that stumbles are responsible for up to 53% of falls in adults [33]. Our results demonstrated that a history of stumbles increased the risk of falling by up to 5.66 times compared to those without a history of stumbles. This highlights the importance of addressing this risk factor during routine clinical assessments with older patients.

**Being a woman increases the risk of falling:** Female sex has been associated with a higher risk of falls, often due to the aggregation of comorbidities such as a high body mass index, vision impairment, depression, joint pain, low physical functioning, and osteoporosis [34]. Furthermore, studies suggest that women may also experience a greater fear of falling, contributing to their increased risk [35,36]. Factors such as nutritional status, the level of independence in daily activities, and the presence of chronic diseases have also been identified as contributors to increased fall risk among women [37]. However, most studies continue to assess the risk of falls in both men and women in association with other pathological and lifestyle factors [38,39]. Our study concludes that being a woman increases the risk of falls by up to 3.55 times compared to men.

**The cases group has greater gait disturbances:** As expected, the cases group had a slower walking speed, an asymmetric arm swing, and a higher unsteadiness during TUG testing. Walking speed has been proposed as an important overall summary of gait function. Slower walking speed may suggest an inability to adapt gait in response to extrinsic factors [40,41]. While arm swing asymmetry is not routinely assessed in most studies, the available evidence, primarily from studies involving young healthy individuals and subjects with pathological conditions like Parkinson’s disease, suggests that arm swing may play a role in balance recovery [42,43,44]. Therefore, reduced arm swing could potentially be associated with an increased risk of falling [45]. These findings underscore the importance of further exploring gait variables in the elderly population without neurodegenerative diseases.

**Visuo-constructional skills and executive functions in cases group:** Considerable evidence has established the relationship between gait and cognition, and how these processes share a functional neuroanatomy. Interestingly, older people with lower levels of education have more risk of cognitive impairment. This has been associated with poor cognitive reserve [46] and worse neuropsychological outcomes that consequently increase the risk of falling [47]. The most common cognitive domain related to gait is executive function (EF). This is a construct that includes attention, sensory integration, and motor planning, which may influence gait dysfunction and the risk of falls.

In our study, the cases group performed worse on the Rey–Osterrieth Complex Figure Test. Previous research has linked visuospatial impairment to falls [48], suggesting that organization and problem-solving skills, necessary for the Rey–Osterrieth Complex Figure Test, could explain why an impairment in this domain supports a higher risk of falls. Additionally, people with falls may have a phonological verbal fluency impairment [49]. Phonological verbal fluency has been associated with executive functions such as response inhibition, conflict monitoring, and working memory. Lower scores on executive tests can lead to lower processing speeds and can compromise the ability to adapt to gait disturbances [50]. Specifically, subjects with executive deficits at baseline are more likely to become recurrent fallers after 2 years [51]. Furthermore, a lower performance on executive function tasks has been observed in people with SMCs. Executive dysfunction has been related to impaired balance and poor recovery from extrinsic factors [17], potentially leading to a higher risk of falling. Although our study did not find differences in other cognitive domains between cases and control groups, we suggest that visuospatial skills and executive functions should be integrated into fall risk screening protocols. The findings and their implications should be discussed in the broadest context possible. Future research directions may also be highlighted.

**Limitations:** However, the limitations of the study design oblige cautious interpretation of the results. The case–control design limits the ability to establish causal relationships. Additionally, the study only evaluated participants who attended the neuropsychological service, potentially limiting generalizability to the broader, older adult population in the community. It is important to note that this study did not collect data on self-reported pain or a history of arthritis, which are also known risk factors for falls in older adults. Another limitation of this study is the use of unsteadiness during TUG testing as a subjective measure, evaluated based on the presence or absence of three characteristics: slow pace, loss of balance, and reduced or absent arm swing, as observed by the evaluator. Also, the use of “estimated walking speed,” derived from TUG testing, is a limitation of this study as it lacks validation as a reliable method for assessing gait speed. The reliability and validity of these measures have not been established, which should be considered when interpreting the findings.

Future studies should address these limitations by employing larger, longitudinal designs to follow up on the cognitive and motor decline associated with stumbles, falls, and “fallers” in older adults.

## 5. Conclusions

Finally, female sex, older age, arterial hypertension, and a history of stumbles were identified as significant risk factors for falls in older adults. Additionally, factors such as asymmetrical arm swing, slow TUG test performance, reduced phonemic verbal fluency, and visuospatial impairment should receive particular attention during clinical assessments of fall risk. These findings emphasize the need for a multidisciplinary approach that integrates cognitive and motor evaluations in assessing and managing fall risk, especially in individuals with subjective memory complaints and mild cognitive impairment.

The results of this study carry important implications for the clinical evaluation of older adults, particularly in neuropsychological consultations. Falls in later adulthood often contribute to declines in functional independence and quality of life and may correlate with mild cognitive impairment. This study is novel in highlighting the importance of a comprehensive clinical approach that includes the assessment of stumbles and falls, as well as the interplay between cognition and gait. A proposed protocol should integrate the cognitive screening of executive functions and visuospatial skills, thereby promoting personalized strategies to prevent mobility decline in adults over 50 years of age.

## Figures and Tables

**Figure 1 ijerph-21-01683-f001:**
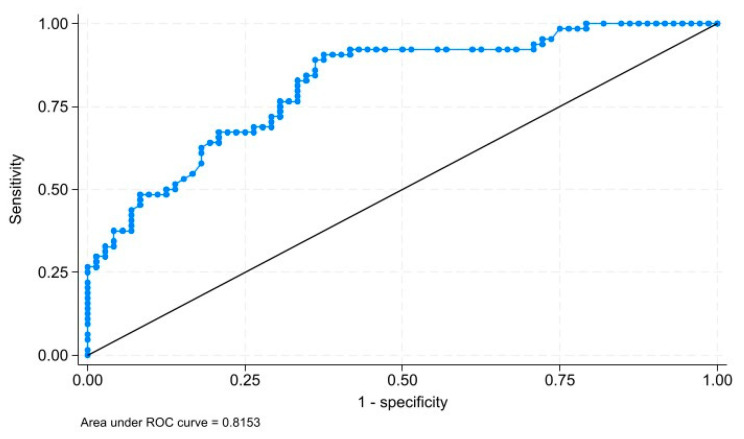
The area under the ROC curve (0.8153) to discriminate patients with mild cognitive impairment to probability of falling of the adjusted model.

**Table 1 ijerph-21-01683-t001:** Demographic characteristics and clinical history of the sample.

Characteristics	Participants (*n* = 136)	Cases (Fallers) (64)	Controls (Non-Fallers) (72)	*p* Value
Gender (*n*, %) ^a^				0.057
Female	82 (60.29)	44 (32.35)	38 (27.94)	
Male	54 (39.71)	20 (14.71)	34 (25.00)	
Age categories (*n*, %) ^b^				0.059
<65 years	45 (33.09)	16 (25)	29 (40.28)	
≥65 years	91 (66.91)	48 (75)	43 (59.72)	
Age, years (mean ± SD) ^c^	67.47 ± 8.24	70.76 ± 7.31	64.55 ± 7.95	<0.001 *
Educational attainment (*n*, %) ^a^				0.027 *
Elementary school	10 (7.35)	8 (5.88)	2 (1.47)	
High school	37 (27.21)	19 (13.97)	18 (13.24)	
Technical–Professional	69 (50.74)	25 (18.38)	44 (32.35)	
Post-graduate	20 (14.71)	12 (8.82)	8 (5.88)	
Pathological history (*n*, %) ^a^				
Hypertension	62 (45.93)	40 (29.63)	22 (16.30)	<0.001 *
Diabetes	21 (15.56)	11 (8.15)	10 (7.41)	0.619
Dyslipidemia	44 (32.59)	25 (18.52)	19 (14.07)	0.128
Hypothyroidism	46 (34.07)	24 (17.78)	22 (16.30)	0.425
Coronary heart disease	9 (6.67)	5 (3.70)	4 (2.96)	0.735
Orthopedic	61 (44.85)	34 (25.00)	27 (19.85)	0.067
Vertigo	21 (15.44)	11 (8.09)	10 (7.35)	0.595
Instability	36 (26.47)	19 (13.97)	17 (12.50)	0.423
Stumble	24 (17.65)	19 (13.97)	5 (3.68)	0.001 *
Consumption of antihypertensives	29 (24.69)	17 (68)	12 (27.91)	0.001 *
Number of antihypertensives consumed (*n*, %) ^a^				0.007
None	76 (56.30)	27 (42.19)	49 (69.01)	
One	37 (27.41)	24 (37.50)	13 (18.31)	
Two or more	23 (16.30)	13 (20.31)	10 (12.68)	
Consumption of medications for diabetes (*n*, %) ^a^	8 (11.76)	4 (16)	4 (9.30)	0.453

^a^ Pearson’s chi-square test. ^b^ Fisher’s exact test. ^c^ Continuous variables (mean, SD) were tested for between-group differences using the Student’s *t* test. * *p* < 0.05

**Table 2 ijerph-21-01683-t002:** The gait and cognitive performance of the sample.

Characteristics	Participants (136)	Cases (Fallers) (64)	Controls (Non-Fallers) (72)	*p* Value
Gait Characteristics				
Single-task TUG (sec) (mean ± SD) ^a^	13.68 ± 3.16	14.43 ± 3.48	13.01 ± 2.69	0.008 *
Dual-task TUG (sec) (mean ± SD) ^a^	16.05 ± 4.61	17.24 ± 5.07	15.00 ± 3.90	0.004 *
Estimated single-task walking speed (m/s) (mean ± SD) ^a^	0.45 ± 0.09	0.43 ± 0.09	0.47 ± 0.08	0.009 *
Estimated dual-task walking speed (m/s) (mean ± SD) ^a^	0.40 ± 0.10	0.37 ± 0.10	0.42 ± 0.10	0.005 *
Cognitive cost (mean ± SD) ^a^	12.96 ± 11.02	14.58 ± 9.76	11.51 ± 11.91	0.105
Unsteadiness during TUG testing, *n* (%) ^b^	17 (12.50)	13 (9.56)	4 (2.94)	0.009 *
Instability, *n* (%) ^c^	18 (13.24)	11 (8.09)	7 (5.15)	0.200
Slow TUG test performance, *n* (%) ^b^	43 (31.62)	26 (19.12)	17 (12.50)	0.033 *
Asymmetric arm swing, *n* (%) ^b^	45 (33.09)	27 (19.85)	18 (13.24)	0.033 *
Neuropsychological Assessment				
Global Cognition				
MoCA test score (median–IQR) ^d^	23 (21–25)	23 (20–25)	23 (21–26)	0.351
MoCA Category, *n* (%)				0.223
Mild cognitive impairment	102 (75)	50 (78.13)	52 (72.22)	
Normal	34 (25)	14 (21.88)	20 (27.78)	
Visuo-Constructional Skills				
Block design score (median–IQR) ^d^	8 (6–8)	8 (6–9)	7 (6–8)	0.259
Rey–Osterrieth figure copy (median–IQR) ^d^	31 (28–34)	30 (26–32)	32 (29–34)	0.047 *
Semantic Verbal Fluency				
Naming animals (mean ± SD) ^a^	16.53 ± 4.50	15.79 ± 4.28	17.19 ± 4.62	0.071
Naming fruits (mean ± SD) ^a^	12.70 ± 3.47	12.20 ± 3.43	13.15 ± 3.46	0.111
Executive Functions				
Phonological Verbal Fluency				
Number of words beginning with F (mean ± SD) ^a^	11.21 ± 3.49	10.53 ± 3.62	11.81 ± 3.29	0.043 *
Number of words beginning with A (mean ± SD) ^a^	11.03 ± 3.89	10.54 ± 3.65	11.47 ± 4.08	0.168
Number of words beginning with S (mean ± SD) ^a^	10.88 ± 4.05	10.14 ± 3.91	11.54 ± 4.08	0.043 *
Total, Frontal Assessment Battery (FAB) score (mean ± SD) ^a^	13.33 ± 2.56	12.70 ± 2.52	13.90 ± 2.47	0.060
Working memory index (mean ± SD) ^a^	88.72 ± 10.09	88.30 ± 10.09	89.08 ± 9.20	0.658
Non-verbal and Verbal MemoryRey–Osterrieth figure recall (mean ± SD) ^a^	12.05 ± 6.74	11.45 ± 6.66	12.59 ± 6.81	0.325
Auditory memory index (mean ± SD) ^a^	93.97 ± 17.51	92.76 ± 15.48	95.05 ± 19.18	0.448

^a^ Continuous variables (mean, SD) were tested for differences between the two groups using the Student’s *t* test. ^b^ Dichotomous variables (n, %) were tested using the Pearson’s chi-square test. ^c^ Fisher’s exact test. ^d^ Continuous variables (median, RIC) were tested for differences between the two groups using the Mann–Whitney U test. * *p* < 0.05.

**Table 3 ijerph-21-01683-t003:** Effect of age interval and frequency of risk factors.

Characteristics		Age (Years)		*p* Value
	Total	<65	≥65	
Falls				
Yes (Case)	64	16	48	0.059 ^a^
No (Control)	72	29	43	
Total	136	45	91	
Frequency	47.06%	35.56%	52.75%	
OR		1.00	2.02 (0.96–4.22)	
Pathological history				
Arterial hypertension				0.121 ^a^
Yes	62	16	46	
No	74	29	45	
Total	136	45	91	
Frequency	45.59%	35.56%	50.55%	
OR		1.00	1.78 (0.85–3.74)	
Diabetes				
Yes	21	7	14	0.937 ^a^
No	115	38	77	
Total	136	45	91	
Frequency	15.44%	15.56%	15.38%	
OR		1.00	0.96 (0.35–2.58)	
Dyslipidemia				
Yes	91	11	33	0.191 ^a^
No	45	34	58	
Total	136	45	91	
Frequency	66.91%	24.44%	36.26%	
OR		1.00	1.70 (0.76–3.81)	
Hypothyroidism				
Yes	46	10	36	0.053 ^a^
No	90	35	55	
Total	136	45	91	
Frequency	33.82%	22.22%	39.56%	
OR		1.00	2.22 (0.97–5.05)	
Coronary heart disease				
Yes	126	2	84	0.390 ^b^
No	10	43	7	
Total	136	45	91	
Frequency	92.65%	4.44%	92.31%	
OR		1.00	1.75 (0.34–8.79)	
Orthopedic				
Yes	61	22	39	0.506 ^a^
No	75	23	52	
Total	136	45	91	
Frequency	44.85%	48.89%	42.86%	
OR		1.00	0.78 (0.38–1.60)	
Vertigo				
Yes	115	8	13	0.596 ^a^
No	21	37	78	
Total	136	45	91	
Frequency	84.56%	17.78%	14.29%	
OR		1.00	0.77 (0.29–2.02)	
Instability				
Yes	36	10	65	0.430 ^a^
No	100	35	26	
Total	136	35	91	
Frequency	26.47%	28.57%	71.43%	
OR		1.00	1.4 (0.60–3.23)	
Stumble				
Yes	24	4	20	0.060 ^a^
No	112	41	71	
Total	136	45	91	
Frequency	17.65%	8.89%	21.98%	
OR		1.00	2.88 (0.92–9.03)	
Single-task TUG (sec) (mean ± SD) ^a^	13.68 ± 3.16	12.65 ± 2.24	14.19 ± 3.42	0.007 ^c,^*
Dual-task TUG (sec) (mean ± SD) ^a^	16.05 ± 4.61	14.55 ± 3.81	16.80 ± 4.80	0.007 ^c,^*
Single-task walking speed. (m/s) (mean ± SD) ^a^	0.45 ± 0.09	0.48 ± 0.08	0.44 ± 0.09	0.011 ^c,^*
Dual-task walking speed. (m/s) (mean ± SD) ^a^	0.40 ± 0.10	0.43 ± 0.10	0.38 ± 0.09	0.004 ^c,^*
Cognitive cost (mean ± SD) ^a^	13.32 ± 10.50	11.74 ± 10.30	14.10 ± 10.57	0.218 ^c^

^a^ Continuous variables (mean, SD) were tested for differences between the two groups using the Student’s *t* test. ^b^ Dichotomous variables (n, %) were tested using the Pearson’s chi-square test. ^c^ Fisher’s exact test. * *p* < 0.05

**Table 4 ijerph-21-01683-t004:** Logistic regression analysis: factors associated with falls.

Characteristics	Participants (136 (100%))	Patients (64 (47.06%))	Controls (72 (52.94%))	Basic Model		Final Model	
				OR (IC 95%)	*p* Value	OR (IC 95%)	*p* Value
**Gender. *n* (%)**							
Female	82 (60.29)	44 (68.75)	38 (52.78)	1.96 (0.97–3.97)	0.059	3.55 (1.44–8.76)	0.006
Age (Years)	67.47 ± 8.24	70.76 ± 7.31	64.55 ± 7.95	1.11 (1.05–1.17)	<0.001	1.08 (1.02–1.14)	0.006
Postgraduate educational level (Years)	20 (14.71)	12 (8.82)	8 (5.88)	1.46 (0.49–4.36)	0.493	0.36 (0.15–0.85)	0.021
**Pathological history. *n* (%)**							
Arterial hypertension	62 (45.93)	40 (62.50)	22 (30.99)	3.71 (1.81–7.57)	<0.001	3.33 (1.43–7.74)	0.005
**Gait characteristics. *n* (%)**							
Stumbles	24 (17.65)	19 (29.69)	5 (6.94)	5.65 (1.96–16.25)	0.001	5.66 (1.62–19.75)	0.002

## Data Availability

All data generated or analyzed during this study are included in this article. Further inquiries can be directed to the corresponding author.

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
