# Peer review of "Stumbles, Gait, and Cognition: Risk Factors Associated with Falls in Older Adults with Subjective Memory Complaints"

_ijerph, 2024, doi:10.3390/ijerph21121683_

Round 1
Reviewer 1 Report
Comments and Suggestions for Authors
The work "Stumbles, gait, and cognition: risk factors associated with falls in older adults with subjective memory complaints" is very high quality. The authors did a good job. Although I discovered what was left from the draft.
These are lines 122, 126, 128, 132, 133. The numbers are probably cited sources that are not properly labeled.
In addition, there is a parameter in the tables - Dual walking velocity. But he is not mentioned in the Materials. I don't remember such a parameter and I think it needs to be explained what it is.
Among other things, these are only the authors’ wishes for further research touse, in addition to clinical scales, instrumental methods for studying walking
and balance in an upright stance or other positions and locomotion. Now there
are very accessible methods. In any case, their sensitivity is not comparable
to clinical scales.
Author Response
The work "Stumbles, gait, and cognition: risk factors associated with falls in older adults with subjective memory complaints" is very high quality. The authors did a good job.
- Although I discovered what was left from the draft. These are lines 122, 126, 128, 132, 133. The numbers are probably cited sources that are not properly labeled.
- ANSWER: Thank you for the review and recommendations. The lines have been revised according to your suggestion; sources are now properly cited.
In addition, there is a parameter in the tables - Dual walking velocity. But he is not mentioned in the Materials. I don't remember such a parameter and I think it needs to be explained what it is.
- ANSWER: Thank you for the review and recommendations. The parameter was revised according to the suggestion; the definition has been included in Materials and Methods. Line 147-148.
- Among other things, these are only the authors’ wishes for further research to use, in addition to clinical scales, instrumental methods for studying walking and balance in an upright stance or other positions and locomotion. Now there are very accessible methods. In any case, their sensitivity is not comparable to clinical scales.
- ANSWER: Thank you for the review and recommendations.
Reviewer 2 Report
Comments and Suggestions for Authors
The article entitled “Stumbles, gait, and cognition: risk factors associated with falls in older adults with subjective memory complaints” aims to explore the risk factors associated with falls in adults with subjective memory complaints (SMCs) and mild cognitive impairment (MCI). A total of 136 patients were enrolled in this study. Neuropsychological assessment was done for all the patients, for stumble, instability, arm swing symmetry, risk of falls, cognitive cost, walking speed. Other parameters considered for this study were sex, age, educational attainment, pathological history like, diabetes, hypertension, etc. The study concluded that older older adults with sensory motor cognitive impairments who experienced falls were more likely to be female, have hypertension, and have a history of stumbling.They also showed lower verbal fluency and impaired visuoconstructional abilities. The study highlights the importance of monitoring cognitive and motor functions in older adults to identify risk factors for falls and implement prevention strategies.
The article is generally well written, and methods provide sufficient details. The table in this article makes the data easy to understand and interpret the outcomes.
After a careful reading, I did not find any drawbacks or shortcomings that required major revisions.
In conclusion, this article is ready for the publication.
Typo Error: Line 24 : Visuoconstruccional
Author Response
REVIEWER 2
The article entitled “Stumbles, gait, and cognition: risk factors associated with falls in older adults with subjective memory complaints” aims to explore the risk factors associated with falls in adults with subjective memory complaints (SMCs) and mild cognitive impairment (MCI). A total of 136 patients were enrolled in this study. Neuropsychological assessment was done for all the patients, for stumble, instability, arm swing symmetry, risk of falls, cognitive cost, walking speed. Other parameters considered for this study were sex, age, educational attainment, pathological history like, diabetes, hypertension, etc. The study concluded that older adults with sensory motor cognitive impairments who experienced falls were more likely to be female, have hypertension, and have a history of stumbling. They also showed lower verbal fluency and impaired visuoconstructional abilities. The study highlights the importance of monitoring cognitive and motor functions in older adults to identify risk factors for falls and implement prevention strategies.
The article is generally well written, and methods provide sufficient details. The table in this article makes the data easy to understand and interpret the outcomes.
After a careful reading, I did not find any drawbacks or shortcomings that required major revisions.
In conclusion, this article is ready for the publication.
- Typo Error: Line 24: Visuoconstruccional
ANSWER: Thank you for the review and recommendations. The line has been revised according to your suggestion; the error has been corrected.
Reviewer 3 Report
Comments and Suggestions for Authors
The article aims to: This study investigated the risk factors associated with falls in a Colombian population aged over 50 years with SMCs.
The authors are asked to address the following points:
- The research objective stated in the discussion section is more precise. It is recommended to use it consistently across all relevant sections.
- It is recommended to list all the inclusion criteria for the sample for better reader understanding.
- It is unclear whether the sample is representative of the population. If not, this should be declared as a limitation of the study, explaining the effects this characteristic may have on the analysis of results.
- It is advisable to determine the statistical power and effect size (G*power software can be used for this).
- In the statistical analysis, it is not clear whether there is a normal distribution of the data, as both parametric and non-parametric statistics are used by the authors.
- The age stratification (Line: 177-178) should be declared in the population subsection, specifying all characteristics of each independent group.
- It is recommended to introduce the study’s limitations in the last paragraph at the end of the discussion section.
- The conclusions should be more concise and brief.
Consult with an English language specialist
Author Response
The authors are asked to address the following points:
- The research objective stated in the discussion section is more precise. It is recommended to use it consistently across all relevant sections.
- ANSWER: Thank you for the review and recommendations. The line has been revised according to the suggestion, objective was standardized. “This study aims to investigate the risk factors associated with falls in a Colombian population aged over 50 years with SM”. Line 80-81.
- It is recommended to list all the inclusion criteria for the sample for better reader understanding.
- ANSWER: Thank you for the review and recommendations. The inclusion criteria for the sample were added. Line 100-101.
- It is unclear whether the sample is representative of the population. If not, this should be declared as a limitation of the study, explaining the effects this characteristic may have on the analysis of results.
- ANSWER: Thank you for reviewing this paper and the recommendations. The limitation regarding population size has been evaluated, since the study power was greater than 80%, we consider that the sample size obtained for this study was adequate, and therefore, the results are reliable.
It is advisable to determine the statistical power and effect size (G*power software can be used for this).
- ANSWER: Thank you for reviewing this paper and the recommendations. The power and effect size has been calculated, yielding a power of 87% and effect size (R² = 0.0034). Lines 293-303.
- In the statistical analysis, it is not clear whether there is a normal distribution of the data, as both parametric and non-parametric statistics are used by the authors.
- ANSWER: Thank you for reviewing this paper and the recommendations. The distribution of numeric variables was analyzed individually; variables with a parametric distribution are reported as the mean and standard deviation, while those with a non-parametric distribution are reported as the median and interquartile range. Lines 184–187.-187.
- The age stratification (Line: 177-178) should be declared in the population subsection, specifying all characteristics of each independent group
- ANSWER: Thank you for reviewing this paper and the recommendations. Age was stratified into two groups: < 65 years and > 65 years of age. The cut-off was established based on the WHO definition of older adult. No characteristics other than those cited in the inclusion and exclusion criteria were considered.
- It is recommended to introduce the study’s limitations in the last paragraph at the end of the discussion section.
- ANSWER: Thank you for reviewing this paper and the recommendations. The study’s limitations have been included in the last paragraph at the end of the Discussion section. Lines 391–400.
- The conclusions should be more concise and brief.
- ANSWER: Thank you for reviewing this paper and the recommendations. The conclusion has been revised.
Reviewer 4 Report
Comments and Suggestions for Authors
Thank you for the opportunity to review this paper "Stumbles, gait, and cognition: risk factors associated with falls in older adults with subjective memory complaints".
It is an increasingly topical subject, given the ageing of the population.
I would like to make some suggestions/corrections in different sections of the paper to improve it.
Abstract
In the abstract, the key words should be repeated with the words that are already in the title to promote greater dissemination of the paper, they should correct this aspect and not number them.
Materials and Methods
It's not clear how they accessed all the individuals' information, as this study is retrospective. This issue should be made clearer to the reader.
Discussion
Since this is a retrospective study that took place between 2018 and 2022, it would be important to understand and discuss what is being carried out today to reduce the impact of the outcomes studied. It is important for this study to already have current data on the impact of the outcomes studied.
The discussion should also be reformulated in this direction.
Conclusions
The conclusion should only respond to the objectives of the study.
The study's limitations and future suggestions should be included in the discussion.
References
They should review the references, as there are many studies that are more than 15 years old.
Author Response
Thank you for the opportunity to review this paper "Stumbles, gait, and cognition: risk factors associated with falls in older adults with subjective memory complaints".
It is an increasingly topical subject, given the ageing of the population.
I would like to make some suggestions/corrections in different sections of the paper to improve it.
Abstract
- In the abstract, the keywords should be repeated with the words that are already in the title to promote greater dissemination of the paper, they should correct this aspect and not number them.
- ANSWER: Thank you for reviewing this paper and the recommendations. Keywords have been revised, modified, and are no longer numbered.
- Materials and Methods, it is not clear how they accessed all the individuals' information, as this study is retrospective. This issue should be made clearer to the reader.
- ANSWER: Thank you for reviewing this paper and the recommendations. Data was collected from clinical records. Line 93.
- Discussion, since this is a retrospective study that took place between 2018 and 2022, it would be important to understand and discuss what is being carried out today to reduce the impact of the outcomes studied. It is important for this study to already have current data on the impact of the outcomes studied. The discussion should also be reformulated in this direction.
- ANSWER: Thank you for reviewing this paper and the recommendations. Unfortunately, we did not have prior information on these outcomes, nor had we implemented any strategies specifically addressing the identified risk factors.
- Conclusions, the conclusion should only respond to the objectives of the study. The study's limitations and future suggestions should be included in the discussion.
- ANSWER: Thank you for reviewing this paper and the recommendations. The conclusion has been revised, and the study’s limitations have been added to the last paragraph of the Discussion section. Lines 391–400.
- References, they should review the references, as there are many studies that are more than 15 years old.
ANSWER: Thank you for reviewing this paper and the recommendations. In this case, most of the references are from recent studies that update the field’s latest advancements, while older studies provide foundational context.
Round 2
Reviewer 4 Report
Comments and Suggestions for Authors
Thanks again for the opportunity to review the paper "Stumbles, gait, and cognition: risk factors associated with falls in older adults with subjective memory complaints"
The present reformulations are clear and help to clarify the paper.
I make a few suggestions to improve the work.
The key words keep repeating themselves with the words in the title, you should replace them with others.
The bibliographical references at the end of the document do not follow the journal's guidelines; they should all be revised.
Author Response
- The key words keep repeating themselves with the words in the title, you should replace them with others.
- ANSWER: Thank you for the review and recommendations. The key words have changed: Aging, Memory disorder, falls, sex.
- The bibliographical references at the end of the document do not follow the journal's guidelines; they should all be revised.
- ANSWER: Thank you for the review and recommendations. The references have been changed for the style recommended in the Author Guidelines Journal web page. All references have been checked carefully in the text.